# Quantum phase transition in Ξ-configuration Tavis-Cummings model driven by an electromagnetic field for a finite number of atoms

A. Alvarado-Sánchez[1*] and R. López-Peña[1†]

**1** Instituto de Ciencias Nucleares, Universidad Nacional Autónoma de México, Apdo. Postal 70-543, 04510, Cd.Mx., México.

* anahi_a.s@ciencias.unam.mx , † lopez@nucleares.unam.mx

## Abstract

The quantum phase transitions of a three-level Tavis-Cummings model in the Ξ-configuration driven by one-mode electromagnetic field in a QED cavity are studied for $N_a = 1$, 2, and 4 systems. Although the groundstate expectation value of usual observables, such as the number of photons and the population at the lowest and most excited levels, can be used to determine the quantum phase transitions of the system in the parameter space, it is shown that expectation values of quadratic Casimir invariants of the subsystems show more details, which are corroborated using the fidelity between neighbouring states. Additionally the entropy of entanglement is calculated to determine the correlations between matter and radiation. All the calculations show that increasing the intensity of the driving electromagnetic field shrinks the region in the parameter space where matter and radiation are decoupled until it dissapears.

# 1   Introduction

Understanding the interaction between matter and electromagnetic radiation at classical and quantum levels has been of fundamental importance in physics and more nowadays due to its potential applications in quantum information theory. To study these systems in quantum optics, Dicke [1] introduced a model in which matter is described as a set of two-level atoms and light as a quantized electromagnetic field inside a cavity. This model transitions from a normal mode, where the $N_a$ atoms radiate individually, to a superradiant phase, where the system's radiation goes proportional to $N_a{}^2$ when the field-matter intensity interaction crosses a critical value [2]. The Dicke model presents the problem that the number of states involved in the groundstate grows in an ever-increasing rate with the intensity of the field-matter inter- action. If the rotating-wave approximation is applied to the Dicke model, a significantly more amenable system is obtained called the Tavis-Cummings model [3, 4].

Quantum phase transitions (QPTs) are phenomena observed in matter-radiation systems [5– 9]. These transitions occur at zero temperature when some physical parameter varies and passes a critical point, and the system shows a non-analytical behavior associated with a dis- continuity in a groundstate property [10, 11]. The study of QPT's is of great interest in the analysis of a variety of quantum systems, such as gases [12], the Ising model [13], Bose- Einstein condensation in an optical cavity [14–16], superconducting systems [17, 18], cold trapped atoms [19], quantum interference [20,21], resonance fluorescence [22–25], etc. Dis- cernment and classifying the nature of nonequilibrium phase transitions in driven-dissipative systems has been a topic of active research [26, 27].

A modification of the two-level Tavis-Cummings model, where a drive of the electromag- netic (EM) mode is implemented, has been considered [28]. In this article a QPT takes place when the driving is detuned from the energy levels difference and the frequency of the EM mode; this transition appears near the first avoided crossing of the groundstate of the model. The model was modified to include a second drive, this time on an ensemble of two-level systems, such that the critical coupling strength can be reduced to observe the QPT in a real- istic system [29]. Simulating the Tavis-Cummings model with a circuit-QED system has been a focus of research and exploration [30]. The modeling of the matter-field interaction has been extended to consider three-level atoms [31]. The phase transitions of two- and three- level atom systems have been studied thermodynamically using the Green function [32]. The quantum phase transitions for the three-level $\Lambda$-configuration interacting with two EM modes were established using the Holstein-Primakov realization of the su(3) algebra in the thermo- dynamic limit [33], and for all the three-level configurations interacting with one-mode EM field using coherent states [34, 35].

In this paper we will consider a generalisation of the Tavis-Cumming model to three-level systems in the $\Xi$-configuration interacting with one EM mode of in a quantum QED cavity when the system is driven by an external field. The work is organized as follows. Section II contains a description of the generalised Tavis-Cummings model. In section III, the model is restricted to the $\Xi$-configuration, and the section IV show the expectation value of some system observables in the groundstate (the energy, the number of photons, the populations in the lowest and the highest states) are plotted to exhibit the QPTs in the model. Then we show that the expectation value of the Quadratic Casimir Invariants show the QPTs more clearly. In the section V we show that the correlation between radiation and matter also exhibits these QPTs. A summary and some concluding remarks are given in section VI.

## 2  The three-level generalised Tavis-Cummings model

The three-level generalized Tavis-Cummings model for $N_a$ systems interacting with the one-mode electromagnetic field in the $\Xi$-configuration is given by [34, 35]

$$\mathbf{H} = \Omega\, \boldsymbol{\nu} + \sum_{i=1}^{3} \omega_i\, \mathbf{A}_{ii} - \frac{1}{\sqrt{N_a}} \sum_{i<j}^{3} \mu_{ij}\left(\mathbf{A}_{ij}\, \mathbf{a}^{\dagger} + \mathbf{A}_{ji}\, \mathbf{a}\right),\tag{1}$$

where $\hbar = 1$ is considered, $\omega_i$, $i = 1, 2, 3$, denotes the energies of the atomic levels, $\mu_{ij}$ is the intensity of the electromagnetic mode interaction between the atomic levels with energies $\omega_i$ and $\omega_j$, $\mathbf{a}^{\dagger}$ and $\mathbf{a}$ are the photon creation and annihilation operators in the field mode with frequency $\Omega$, $\boldsymbol{\nu} = \mathbf{a}^{\dagger}\mathbf{a}$ denotes the photon number operator, $\mathbf{A}_{ij} = \sum_{n=1}^{N_a} \mathbf{A}_{ij}^{(n)}$, $i, j = 1, 2, 3$, are generators of the U(3) group, which are collective operators which annihilate systems at level $\omega_j$ and create systems at level $\omega_i$ ($\mathbf{A}_{ij}^{(n)}$ annihilates a $n$-th system in level $\omega_j$ and creates a $n$-th system in level $\omega_i$), and obey the commutation relations

$$[\mathbf{A}_{ij}, \mathbf{A}_{kl}] = \delta_{jk}\mathbf{A}_{il} - \delta_{il}\mathbf{A}_{kj}, \quad i, j, k, l = 1, 2, 3.\tag{2}$$

The factor $1/\sqrt{N_a}$ in the interaction term is introduced so that parameters $\mu_{ij}$ are intensive. In reference [28], an external electromagnetic forcing of intensity $\chi$ and frequency $\omega$ is introduced to the Dicke model. We will add a similar term to our Hamiltonian in Eq. (1) but multiplied by $\sqrt{N_a}$ so that $\chi$ is an intensive parameter. The Hamiltonian for the $\Xi$-configuration is

$$
\begin{aligned}
\mathbf{H}_{\Xi} \;=\;& \Omega\, \mathbf{a}^{\dagger}\mathbf{a} + \sum_{j=1}^{3} \omega_j\, \mathbf{A}_{jj}\\
&-\frac{1}{\sqrt{N_a}}\left\{\mu_{12}\left(\mathbf{A}_{12}\, \mathbf{a}^{\dagger} + \mathbf{A}_{21}\, \mathbf{a}\right) + \mu_{23}\left(\mathbf{A}_{23}\, \mathbf{a}^{\dagger} + \mathbf{A}_{32}\, \mathbf{a}\right)\right\}\\
&+\sqrt{N_a}\, \chi\left(e^{-i\omega t}\mathbf{a}^{\dagger} + e^{i\omega t}\mathbf{a}\right).
\end{aligned}
$$

The model considered can be depicted as in Figure (1).

The temporal dependence can be eliminated through the unitary tranformation

$$\mathbf{U}(t) = e^{i\, t\left(\alpha\, \mathbf{a}^{\dagger}\mathbf{a} + \beta\, \mathbf{A}_{11} + \gamma\, \mathbf{A}_{22} + \delta\, \mathbf{A}_{33}\right)},\tag{3}$$

if the real constants $\alpha$, $\beta$, $\gamma$, and $\delta$ are chosen as

$$\alpha = \omega, \quad \beta = \gamma - \omega, \quad \delta = \gamma + \omega.\tag{4}$$

The evolution equation for the transformed state $|\phi(t)\rangle$ is

$$i\frac{d}{dt}|\phi(t)\rangle = \left[\mathbf{U}(t)\mathbf{H}_{\Xi}\mathbf{U}^{\dagger}(t) + \left(i\frac{d}{dt}\mathbf{U}(t)\right)\mathbf{U}^{\dagger}(t)\right]|\phi(t)\rangle.\tag{5}$$

Then the new Hamiltonian is

$$
\begin{aligned}
\mathbf{H}_{\Xi}^{(N)} \;=\;& \mathbf{U}(t)\mathbf{H}_{\Xi}\mathbf{U}^{\dagger}(t) + i\left(\frac{d}{dt}\mathbf{U}(t)\right)\mathbf{U}^{\dagger}(t)\\
=\;& (\Omega - \omega)\, \mathbf{a}^{\dagger}\mathbf{a} + (\omega_1 + \omega - \gamma)\, \mathbf{A}_{11} + (\omega_2 - \gamma)\, \mathbf{A}_{22} + (\omega_3 - \omega - \gamma)\, \mathbf{A}_{33}\\
&-\frac{1}{\sqrt{N_a}}\left\{\mu_{12}\left(\mathbf{A}_{12}\, \mathbf{a}^{\dagger} + \mathbf{A}_{21}\, \mathbf{a}\right) + \mu_{23}\left(\mathbf{A}_{23}\, \mathbf{a}^{\dagger} + \mathbf{A}_{32}\, \mathbf{a}\right)\right\}\\
&+\sqrt{N_a}\, \chi\left(\mathbf{a}^{\dagger} + \mathbf{a}\right).
\end{aligned}\tag{6}
$$

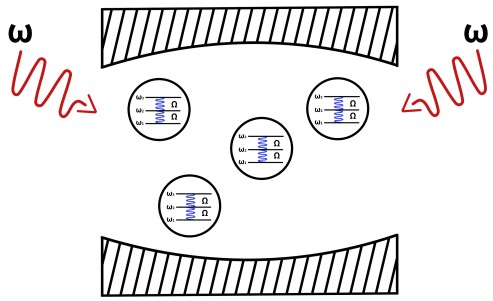

Figure 1: Scheme of $\Xi$-configuration system interacting with one-mode $\Omega$ of a quantum electromagnetic cavity and forced with an electromagnetic field of frequency $\omega$. The interaction with the EM field can induce an atom to go to a higher energy state, absorbing a photon of energy $\Omega$, or pass to a lower energy state, producing a photon of energy $\Omega$. Because an external forcing field of frequency $\omega$ is applied, this new field can also result in this process.

In this Hamiltonian, the EM field frequency of the cavity is decreased by the driving force frequency, and the first and third atomic levels energies, $\omega_1$ and $\omega_3$, approach the middle-level energy $\omega_2$. It is convenient to remove the term $\mathbf{a}^\dagger + \mathbf{a}$ in Eq. (6) to have a better convergence when determining the lowest energy states of the system because increasing the number of photons decreases the energy [28]. To this end we apply the displacement operator $\mathbf{D}(\alpha) = \exp(\alpha\, \mathbf{a}^\dagger - \alpha^*\, \mathbf{a})$ to the previous Hamiltonian. Thus selecting $\alpha = -\sqrt{N_a}\, \chi/(\Omega - \omega)$ the transformed Hamiltonian is

$$
\begin{aligned}
\widetilde{\mathbf{H}}_\Xi &= \mathbf{D}(\alpha)\,\mathbf{H}_\Xi^{(N)}\,\mathbf{D}^\dagger(\alpha) \\
&= (\Omega - \omega)\,\mathbf{a}^\dagger\mathbf{a} + (\omega_1 + \omega - \gamma)\,\mathbf{A}_{11} + (\omega_2 - \gamma)\,\mathbf{A}_{22} + (\omega_3 - \omega - \gamma)\,\mathbf{A}_{33} \\
&\quad - \frac{1}{\sqrt{N_a}}\left\{\mu_{12}\left(\mathbf{A}_{12}\,\mathbf{a}^\dagger + \mathbf{A}_{21}\,\mathbf{a}\right) + \mu_{23}\left(\mathbf{A}_{23}\,\mathbf{a}^\dagger + \mathbf{A}_{32}\,\mathbf{a}\right)\right\} \\
&\quad + \frac{\chi}{\Omega - \omega}\left\{\mu_{13}\left(\mathbf{A}_{13} + \mathbf{A}_{31}\right) + \mu_{23}\left(\mathbf{A}_{23} + \mathbf{A}_{32}\right)\right\} - N_a\frac{\chi^2}{\Omega - \omega}\ ,
\end{aligned}
\tag{7}
$$

with $\gamma$ arbitrary, which we will choose $\gamma = 0$. This Hamiltonian excludes the case when $\omega = \Omega$, but this can be studied using the previous expression (6).

Proceeding in the same manner for the $\Lambda$ and $V$ configurations we obtain

$$
\begin{aligned}
\widetilde{\mathbf{H}}_\Lambda &= (\Omega - \omega)\,\mathbf{a}^\dagger\mathbf{a} + (\omega_1 + \omega - \delta)\,\mathbf{A}_{11} + (\omega_2 + \omega - \delta)\,\mathbf{A}_{22} + (\omega_3 - \delta)\,\mathbf{A}_{33} \\
&\quad - \frac{1}{\sqrt{N_a}}\left\{\mu_{13}\left(\mathbf{A}_{13}\,\mathbf{a}^\dagger + \mathbf{A}_{31}\,\mathbf{a}\right) + \mu_{23}\left(\mathbf{A}_{23}\,\mathbf{a}^\dagger + \mathbf{A}_{32}\,\mathbf{a}\right)\right\} \\
&\quad + \frac{\chi}{\Omega - \omega}\left\{\mu_{13}\left(\mathbf{A}_{13} + \mathbf{A}_{31}\right) + \mu_{23}\left(\mathbf{A}_{23} + \mathbf{A}_{32}\right)\right\} - N_a\frac{\chi^2}{\Omega - \omega}\ ,
\end{aligned}
\tag{8}
$$

$$
\begin{aligned}
\widetilde{\mathbf{H}}_V &= (\Omega - \omega)\,\mathbf{a}^\dagger\mathbf{a} + (\omega_1 - \beta)\,\mathbf{A}_{11} + (\omega_2 + \omega - \beta)\,\mathbf{A}_{22} + (\omega_3 - \omega - \beta)\,\mathbf{A}_{33} \\
&\quad - \frac{1}{\sqrt{N_a}}\left\{\mu_{12}\left(\mathbf{A}_{12}\,\mathbf{a}^\dagger + \mathbf{A}_{21}\,\mathbf{a}\right) + \mu_{13}\left(\mathbf{A}_{13}\,\mathbf{a}^\dagger + \mathbf{A}_{31}\,\mathbf{a}\right)\right\} \\
&\quad + \frac{\chi}{\Omega - \omega}\left\{\mu_{12}\left(\mathbf{A}_{12} + \mathbf{A}_{21}\right) + \mu_{13}\left(\mathbf{A}_{13} + \mathbf{A}_{31}\right)\right\} - N_a\frac{\chi^2}{\Omega - \omega}\ .
\end{aligned}
\tag{9}
$$

In these expressions, $\beta$ and $\delta$ are arbitrary.

## 3  Determination of the groundstate

We shall consider completely symmetric states for the three-level system. This allows us considering the u(3)-algebra Jordan-Schwinger map

$$\mathbf{A}_{jk} = \mathbf{b}_j^\dagger \mathbf{b}_k \,, \quad j, k = 1, 2, 3 \,, \tag{10}$$

where the creation and annihilation operators satisfy the independent harmonic oscillator commutation relations

$$\begin{aligned}
\left[ \mathbf{b}_j, \mathbf{b}_k^\dagger \right] &= \delta_{jk} \mathbf{1} \,, \\
\left[ \mathbf{b}_j, \mathbf{b}_k \right] &= \mathbf{0} = \left[ \mathbf{b}_j^\dagger, \mathbf{b}_k^\dagger \right] \quad j, k = 1, 2, 3 \,.
\end{aligned} \tag{11}$$

Thus a basis for the three-level systems is

$$\{|n_1, n_2, n_3\rangle\} \,, \quad n_j = 0, 1, 2, \dots \,, \quad j = 1, 2, 3 \,. \tag{12}$$

These states are eigenstates of $\mathbf{A}_{jj}$,

$$\mathbf{A}_{jj} |n_1, n_2, n_3\rangle = n_j |n_1, n_2, n_3\rangle \,, \tag{13}$$

and $n_j$ is the number of systems in the j-level. Thus for a system of $N_a$ three-level states the basis

$$\left\{ |N_a, 0, 0\rangle, \ |N_a - 1, 1, 0\rangle, \ |N_a - 1, 0, 1\rangle, \ |N_a - 2, 2, 0\rangle, \ \dots, \ |0, 0, N_a\rangle \right\} \,, \tag{14}$$

contains $(N_a + 1)(N_a + 2)/2$ states. Hence we will use a basis for the matter-radiation system of the form

$$\{ |\nu; n_1, n_2, n_3\rangle \equiv |\nu\rangle \otimes |n_1, n_2, n_3\rangle \} \,. \tag{15}$$

In this expression, states $|\nu\rangle$ are the eigenstates of photon number operator $\nu$. The matrix elements of the Hamiltonian in Eq. (7) in this basis are

$$\begin{aligned}
&\langle \nu; n_1, n_2, n_3 | \widetilde{\mathbf{H}}_\Xi | \nu; n_1', n_2', n_3' \rangle \\
&= \delta_{\nu \nu'} \delta_{n_1 n_1'} \delta_{n_2 n_2'} \delta_{n_3 n_3'} \left[ \left( \Omega - \omega \right) \nu + \left( \omega_1 + \omega \right) n_1 \right. \\
&\quad \left. + \omega_2 n_2 + \left( \omega_3 - \omega \right) n_3 - N_a \frac{\chi^2}{\Omega - \omega} \right] \\
&\quad - \frac{1}{\sqrt{N_a}} \Big[ \mu_{12} \delta_{n_3, n_3'} \Big( \sqrt{\nu'(n_2' + 1) n_1'} \, \delta_{\nu, \nu'-1} \delta_{n_1, n_1'-1} \delta_{n_2, n_2'+1} \\
&\quad + \sqrt{(\nu' + 1)(n_1' + 1) n_2'} \, \delta_{\nu, \nu'+1} \delta_{n_1, n_1'+1} \delta_{n_2, n_2'-1} \Big) \\
&\quad + \mu_{23} \delta_{n_1, n_1'} \Big( \sqrt{\nu'(n_3' + 1) n_2'} \, \delta_{\nu, \nu'-1} \delta_{n_2, n_2'-1} \delta_{n_3, n_3'+1} \\
&\quad + \sqrt{(\nu' + 1)(n_2' + 1) n_3'} \, \delta_{\nu, \nu'+1} \delta_{n_2, n_2'+1} \delta_{n_3, n_3'-1} \Big) \Big] \\
&\quad + \frac{\chi}{\Omega - \omega} \delta_{\nu \nu'} \Big[ \mu_{12} \delta_{n_3, n_3'} \Big( \sqrt{(n_2' + 1) n_1'} \, \delta_{n_2, n_2'+1} \delta_{n_1, n_1'-1} \\
&\quad + \sqrt{(n_1' + 1) n_2'} \, \delta_{n_1, n_1'+1} \delta_{n_2, n_2'-1} \Big) \\
&\quad + \mu_{23} \delta_{n_1, n_1'} \Big( \sqrt{(n_3' + 1) n_2'} \, \delta_{n_3, n_3'+1} \delta_{n_2, n_2'-1} \\
&\quad + \sqrt{(n_2' + 1) n_3'} \, \delta_{n_2, n_2'+1} \delta_{n_3, n_3'-1} \Big) \Big] \,.
\end{aligned}$$

When considering the Tavis-Cumming model generalised to three-level system in the $\Xi$-configuration, the number of excitations operator

$$\mathbf{M} = \mathbf{a}^\dagger \mathbf{a} + \mathbf{A}_{22} + 2 \mathbf{A}_{33} \,, \tag{16}$$

is a conserved quantity. From the relation

$$\mathbf{M}|\nu; n_1, n_2, n_3\rangle = (\nu + n_2 + 2n_3)|\nu; n_1, n_2, n_3\rangle, \tag{17}$$

we observe that $M = \nu + n_2 + 2n_3$ is a constant of this system. This is not the case anymore for Hamiltonian $\widetilde{\mathbf{H}}_\sqsubseteq$ because of the driving force in the model. For example $\mathbf{A}_{i,i+1}$, with $i = 1, 2$; annihilates an atom in level $i+1$ and creates on in level $i$, changing the value of $M$. Thus, to obtain the lowest state of energy of the Hamiltonian, it is natural to consider a basis up to a maximum number of excitations. Comparing the fidelity between the groundstates obtained in the diagonalisation for a consecutive maximum number of excitations, we can obtain a groundstate up to a predetermined accuracy. In our calculations we chose $10^{-15}$, and obtained that $M_{\max} = 90$ was the minimum number of excitations to consider for up to $N_a = 4$ particles. Therefore the groundstate can be written as

$$|\psi\rangle_{gs} = \sum_{M=0}^{M_{\max}} \sum_{\nu, n_3=0} C_{M \nu n_3}^{(gs)} |\nu; N_a - M + \nu + n_3, M - \nu - 2n_3, n_3\rangle, \tag{18}$$

where we must sum over all the values for $\nu$ and $n_3$ such that $N_a - M + \nu + n_3 \geq 0$ and $M - \nu - 2n_3 \geq 0$, and the coefficients $C_{M \nu n_3}^{(gs)}$ are complex numbers. The number of elements of the basis used was 270 for $N_a = 1$, 534 for $N_a = 2$, and 1305 for $N_a = 4$.

# 4 Expectation values in the groundstate

The groundstate energy per particle of Hamiltonian $\widetilde{\mathbf{H}}_\sqsubseteq$ is shown in Figure 2 for $N_a = 4$ particles and various values of driven amplitude $\chi$. It is observed that there is a region around the origin where the surface is approximately flat, i.e., the variation is slight; this region can be identified with the so-called *normal region* (the red region in the plots in Figure 2), where the matter and radiation are decoupled. The curve separating the normal and superradiant regions in the parameter space is called *separatrix*. When the amplitude of $\chi$ increases, the normal region contracts, and the energy values are more negative. This results in a smoother transition to the superradiant region, where the coupling between matter and radiation causes the system to bond more tightly. The groundstate energy behaves similarly for $N_a = 1$ and $N_a = 2$ particles.

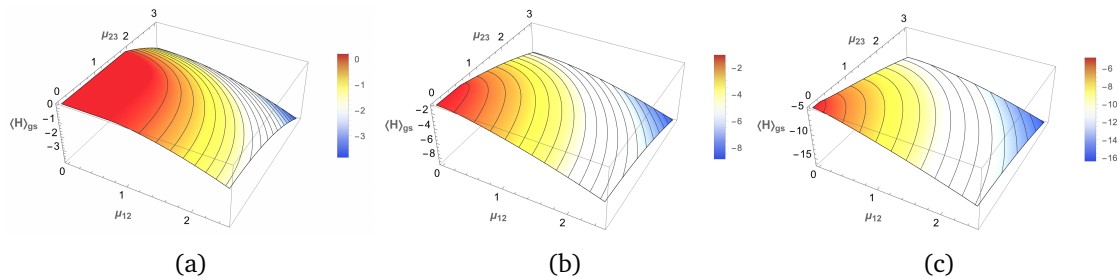

(a)                                       (b)                                       (c)

Figure 2: Groundstate energy for $N_a = 4$ for various values of the amplitude of the driven force $\chi$. The other parameters are $\omega_1 = 0$, $\omega_2 = 1$, $\omega_3 = 2$, $\omega = 0.2$, $\Omega = 1$, and the space was truncated up to $M = 90$ excitations. The values of the driven force amplitude considered are $\chi = 0.01$, $\chi = 1.00$, and $\chi = 2.00$ in columns (a), (b), and (c), respectively. A contraction of the normal region and a smoother change in the surface when passing from the normal region to the superradiant region are observed when the value of drive intebnsity $\chi$ is increased.

In Figure 3, the expectation value per particle of the photon number operator in the ground-state $\langle \nu \rangle_{gs}$, for $N_a = 1, 2, 4$ particles, with an external driven field amplitude value of $\chi = 0.01$, are displayed. It is observed that the normal region grows, and the curve that delimits it becomes more blurred. For $\langle \nu \rangle_{gs}$, well-delimited areas are displayed where the number of photons is constant, but this effect smooths when the number of particles increases. Since the number of particles is low ($N_a \leq 4$), the expectation value of the number of excitations is similar to that of the photon number for the plotted parameters.

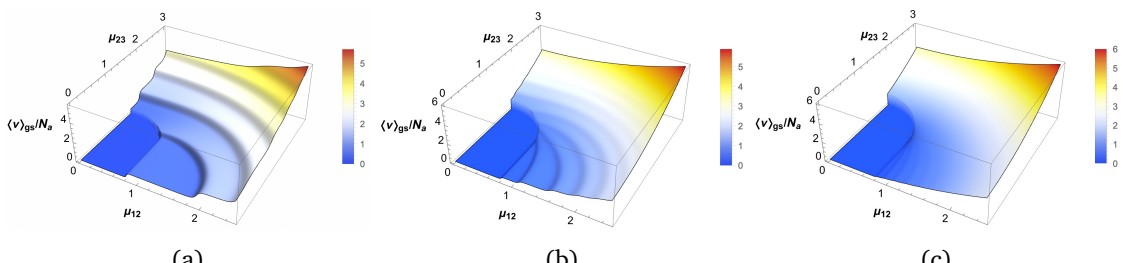

(a)           (b)           (c)

Figure 3: Expectation value of the photon number operator per particle $\langle \nu \rangle_{gs}/N_a$ for $N_a = 1, 2, 4$ in (a), (b) and (c), respectively. The same parameters are used as in Figure 2, but the driven intensity is $\chi = 0.01$ for all the plots. It can be seen that the normal region grows as the number of particles $N_a$ increases. In contrast, the normal region's curve loses definition in the region where the phase transition is of second order for $\chi = 0$.

The expectation value of the population level $i$, $\langle \mathbf{A}_{ii} \rangle_{gs}$, also reflects where are located the normal and superradiant regions in the parameter space.

Figure 4 displays the expectation value of the population in the first level, $\langle \mathbf{A}_{11} \rangle_{gs}$, for $N_a = 1, 2, 4$ (a), (b) and (c) respectively, when the amplitude of the driven field is $\chi = 0.01$. An abrupt change is observed when we pass from the normal region to the superradiant one for small values of $\mu_{12}$ and large values $\mu_{23}$. As we increase the number of particles $N_a$, the normal region grows when $\mu_{23}$ increases. A smoother but gradual change is observed in the behavior of the superradiant region when changing $\mu_{12}$. The number of steps in the curve increases in the superradiant region as the number of particles in the system increases. The region with the largest population, i.e., the maximum values of $\langle \mathbf{A}_{11} \rangle_{gs}$, is obtained for values of $\mu_{23}$ larger than $\mu_{12}$. For large values of $\mu_{23}$ when moving in increasing $\mu_{12}$ direction, a remarkably smooth transition is observed, while a very abrupt change is observed moving in $\mu_{23}$ direction with small values of $\mu_{12}$.

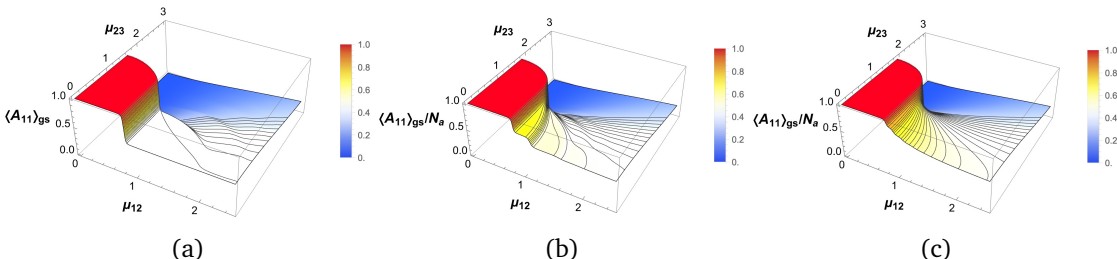

(a)           (b)           (c)

Figure 4: Expectation value of $\langle \mathbf{A}_{11} \rangle_{gs}$ for $N_a = 1, 2, 4$ particles, in (a), (b) and (c), respectively, are shown, keeping $\chi = 0.01$. The other parameters are the same as in Figure 2. The normal region grows when the number of particles is increased. A smoother contour of the normal region is also observed.

155      The expectation value $\langle \mathbf{A}_{22} \rangle_{gs}$ shows a behavior similar to that of $\langle \mathbf{A}_{11} \rangle_{gs}$, showing a well-
156 defined normal region that expanded with an increase in particles. However the normal region
157 for $\langle \mathbf{A}_{11} \rangle_{gs}$ corresponds to the area with the highest population at that level, while the normal
158 region for $\langle \mathbf{A}_{22} \rangle_{gs}$ and $\langle \mathbf{A}_{33} \rangle_{gs}$ corresponds to the area with the lowest population at those
159 levels. The expectation value $\langle \mathbf{A}_{33} \rangle_{gs}$ exhibits a behavior similar to $\langle \mathbf{A}_{22} \rangle_{gs}$, but fails in de-
160 termining the phase transition in the region where the model with $\chi = 0$ has a second order
161 phase. This is because the interaction between the first and second levels dominates around
this neighborhood. Figure 6 illustrates the expectation value of the photon number operator,

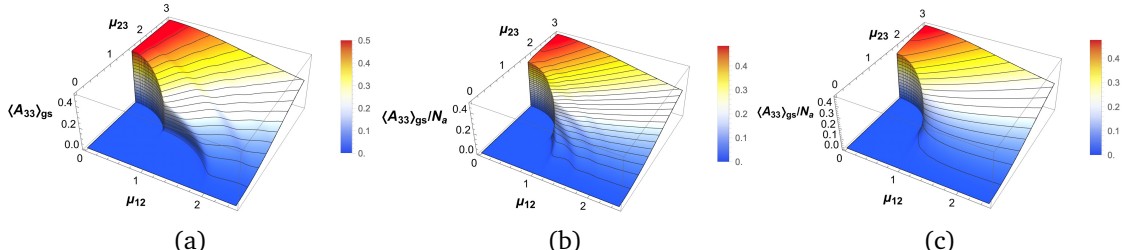

(a)            (b)            (c)

Figure 5: Expectation value of $\langle \mathbf{A}_{33} \rangle_{gs}$ for $N_a = 1, 2, 4$ particles, in (a), (b) and (c),
respectively, are shown, keeping $\chi = 0.01$. The other parameters are the same as
in Figure 2. The separatrix from the normal region to the superradiant region that
we observe in the plots for the expectation values of the population in the first and
second levels is not detected because the interaction between them is so strong that
the influence of the third level is negligible.

162
163 per particle, for the groundstate $\langle \boldsymbol{\nu} \rangle_{gs}$, for $N_a = 4$, when $\chi = 0.3$, $\chi = 0.6$ and $\chi = 1$, in
164 columns (a), (b) and (c), respectively. For $\chi = 2$, results were found close to the results for
165 $\chi = 1$. The deep blue region in the plots indicates the normal region, where the mean value
166 of the photon number is nearly zero. The normal region contracts with increasing external
167 forcing $\chi$, and the curve that delimits the normal region becomes less defined. The minimum
168 values for $\langle \boldsymbol{\nu} \rangle_{gs}$ are in the normal region and the maximum values are found for high values
of $\mu_{12}$ and $\mu_{23}$. Similar results for $N_a = 1, 2$ were obtained. Figure 7 displays the expectation

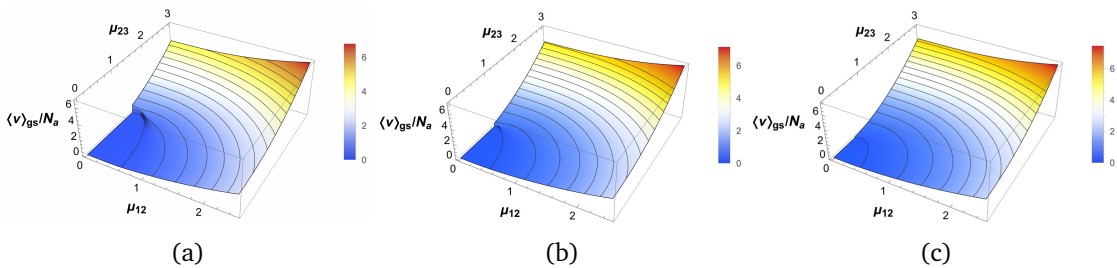

(a)            (b)            (c)

Figure 6: The expectation value of the number of photons per particle $\langle \boldsymbol{\nu} \rangle_{gs}/N_a$ are
shown for $N_a = 4$ and $\chi = 0.3, 0.6, 1$ in columns (a), (b) and (c), respectively. The
parameters used for these calculations are the same as in Figure 2. The normal region
area is smaller when $\chi$ increases, although this fact is unclear in the last plot.

169
170 value of $\langle \mathbf{A}_{11} \rangle_{gs}$ for $N_a = 4$, with an external field amplitude value of $\chi = 0.3, 0.6, 1$ shown
171 in (a), (b) and (c) respectively. For values larger than $\mu_{12}$, there is an observed normal region
172 with an abrupt and well-defined change. The normal region continuously decreases towards
173 $\mu_{12}$ and $\mu_{23}$ as the value of $\chi$ increases. A smoother change in the behavior of the superra-
174 diant region is observed, with an analogous behavior for the number of particles $N_a = 1, 2$.
175 The maximum values of $\langle \mathbf{A}_{11} \rangle_{gs}$ are found for small values of $\mu_{12}$ and $\mu_{23}$, which correspond
176 to the region with the largest population. A slight halo of higher values is also observed in the

contour of the normal region. For large values of $\mu_{23}$ moving in $\mu_{12}$, very smooth transitions are observed, while very abrupt changes are observed moving in $\mu_{23}$ for small values in $\mu_{12}$. Similar results are obtained for the case of 1 and 2 particles. The plots for $\langle \mathbf{A}_{22} \rangle_{gs}$, with $\chi =$

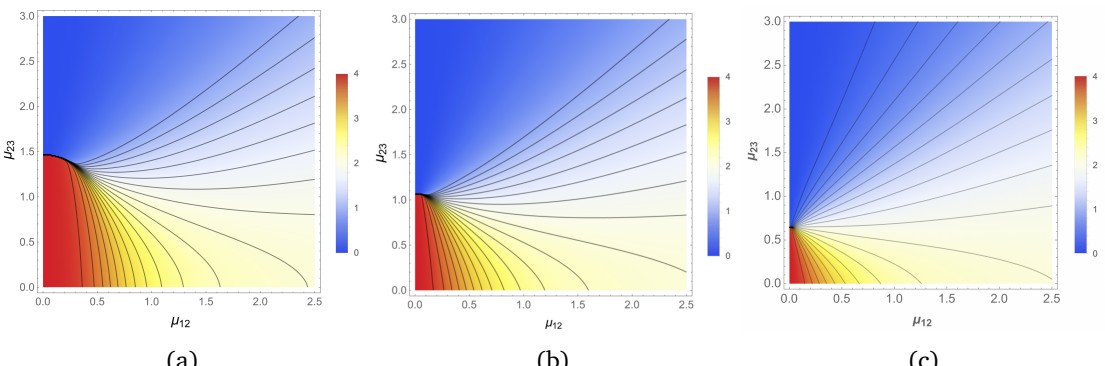

(a)            (b)            (c)

Figure 7: In the visualization, we can see the expectation value $\langle \mathbf{A}_{11} \rangle_{gs}$ in a 2D view for $N_a = 4$ particles, for $\chi = 0.3, 0.6, 1$ in (a), (b) and (c) respectively, per particle unit. The parameters used here are the same as in Figure 2. We observe that as the value of the external field $\chi$ increases, a contraction is observed in the normal region.

0.3, 0.6, 1, show similar behaviour of the normal and superradiant regions in the parameter space of the plots for $\langle \mathbf{A}_{11} \rangle_{gs}$ in the sense that when one takes maximum values, the other takes minimum values and vice versa. In both cases, it is noted that the normal region is compressed with an increase in the value of $\chi$.

Due to the behaviour of $\langle \mathbf{A}_{33} \rangle_{gs}$ it is convenient to search for another observables whose expectation value shows the separatrix in the parameter space with more detail. Because in the $\Xi$-configuration the interaction is mediated between levels $1-2$ and $2-3$, it seems appropriate to use operators that relate this subsystems.

When $\mu_{23} = 0$, the system is a forced TC two-level Hamiltonian. When $\mu_{23}$ starts to increase, this behaviour is modified so that previously conserved quantities will change.

To reinforce the results obtained in the calculation of the expected values shown previously, which indicated when a quantum phase transition occurs, the fidelity between neighboring states was calculated [36]. To this effect the fidelity between adjacent states varying $\mu_{12}$ keeping $\mu_{23}$ constant was obtained; the same process was done varying $\mu_{23}$ keeping $\mu_{12}$ constant; and finally the geometric mean of the values was evaluated. Figure 8 displays the overlap calculation. It can be noted that since the states are eigenstates of the entire system, the fidelity is generally close to 1, but presents minima which determine points in the separatrix signaling discontinuous or continuous transitions [37]. We can observe in the image that the normal region, the region in the graphs in deep red, is reduced as the value of the amplitude of the external field increases, in accordance with the previously obtained results.

We have seen that the expectation values in the groundstate of the usual observables of the system have a remarkable behaviour in the regions where a quantum phase transition occurs. This response to the separatrix of the system can be better determined if we use the expectation value of the Casimir operator associated to subsystems associated to levels $1-2$ and $2-3$. The generators relating levels $j$ and $k$, with $\omega_k \geq \omega_j$, give rise to a su(2) algebra

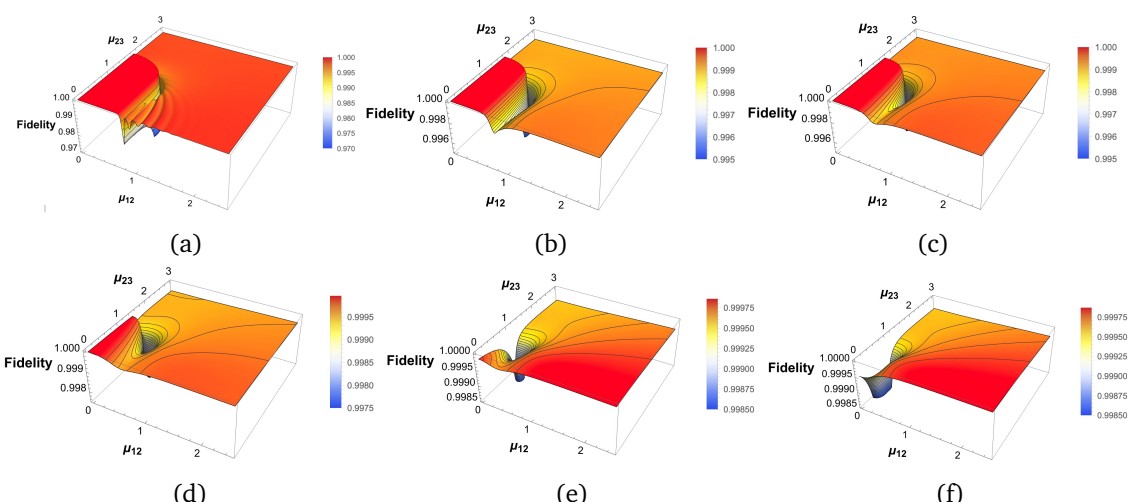

Figure 8: The fidelity shows the results for $N_a = 4$ and $\chi = 0.001, 0.005, 0.01, 0.3, 0.6, 1.0$, in columns (a), (b), (c), (d), (e), and (f), respectively. We can observe that by increasing the value of the amplitude of the external field $\chi$ the normal region that corresponded to small values of $\mu_{12}$ and $\mu_{23}$ becomes compacted until it finally disappears.

making the identification

$$\mathbf{J}_z^{(jk)} \equiv \frac{1}{2}\left(\mathbf{A}_{kk} - \mathbf{A}_{jj}\right) , \quad \mathbf{J}_+^{(jk)} \equiv \mathbf{A}_{kj} , \quad \mathbf{J}_-^{(jk)} \equiv \mathbf{A}_{jk} , \quad j < k . \tag{19}$$

Thus the quadratic Casimir of $i - j$ subsystem is

$$\langle \mathbf{C}_{ij} \rangle_{gs} = \frac{1}{4}\langle \mathbf{A}_{ii}^2 + \mathbf{A}_{jj}^2 - 2\mathbf{A}_{ii}\mathbf{A}_{jj} + 2\mathbf{A}_{ji}\mathbf{A}_{ij} + 2\mathbf{A}_{ij}\mathbf{A}_{ji} \rangle_{gs}. \tag{20}$$

The large values this operator occur when one of the interaction parameters $\mu_{12}$ or $\mu_{23}$ is more important in the interaction than the other.

The subsystem $i - j$ is the main responsible for the behaviour of the total system in the regions where the expectation value of its corresponding Casimir operator $\mathbf{C}_{ij}$ has maximum value.

In Figure 9 the expectation value of the quadratic Casimir invariant of first and second levels $\langle \mathbf{C}_{12} \rangle_{gs}$ is shown (top) and for the second and third levels $\langle \mathbf{C}_{23} \rangle_{gs}$ (bottom) is displayed for $\chi = 0.01$ and $N_a = 1, 2, 4$ in (a), (b) and (c) respectively. These invariants show where each subsystem is more important. For $\langle \mathbf{C}_{12} \rangle_{gs}$, the entire yellow and red region is the most important, corresponding to small values for $\mu_{23}$. On the contrary, the region of minor importance for this subsystem is the deep blue region corresponding to high values of $\mu_{23}$ and small values of $\mu_{12}$. For $\langle \mathbf{C}_{23} \rangle_{gs}$, the region of greatest importance lies in the region of least importance for $\langle \mathbf{C}_{12} \rangle_{gs}$. That is, for high values of $\mu_{23}$ and small values of $\mu_{12}$. For both invariants, it can be seen that for one particle, the system has very well-defined regions for the different values the subsystem takes. These jumps smooth out as the number of particles increases.

The plots for $\langle \mathbf{C}_{12} \rangle_{gs}$ and $\langle \mathbf{C}_{23} \rangle_{gs}$ for $\chi = 0.3$, 0.6 and 1, also show the normal region and the transitions from the normal region to the superradiant region. These results are similar to those obtained for $\langle \mathbf{A}_{11} \rangle_{gs}$. Additionally, it is important to note that for subsystem $1 - 2$, the most significant region corresponds to small values of $\mu_{23}$ and any value of $\mu_{12}$. For the

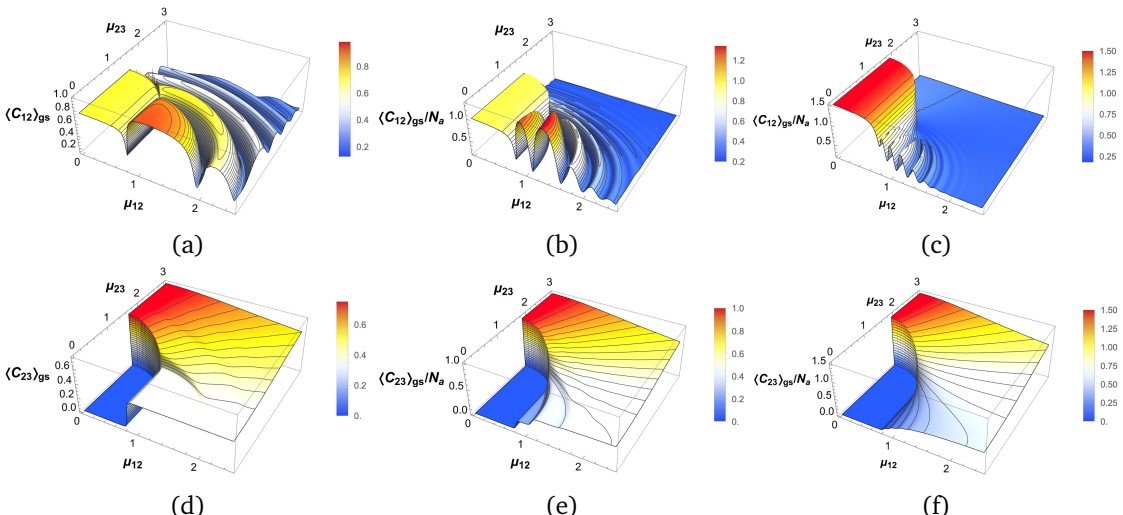

Figure 9: The expectation value of quadratic Casimir operators $\langle \mathbf{C}_{12}\rangle_{gs}$ (top), and $\langle \mathbf{C}_{23}\rangle_{gs}$ (bottom), for $\chi = 0.01$ and $N_a = 1, 2, 4$ ((a), (b) and (c) respectively). The same parameters are used as in Figure 2. These invariants show where each subsystem is more important.

subsystem $2-3$, the large values region is in the area of small values for $\langle \mathbf{C}_{12}\rangle_{gs}$, and viceversa. The plots show that $\langle \mathbf{C}_{12}\rangle_{gs}$ is more useful than $\langle \mathbf{C}_{23}\rangle_{gs}$ to determine the separatrix of the system because also exhibits regions in the superradiant region where there is a change in the groundstate, i.e., the expectation value of $\langle \mathbf{C}_{12}\rangle_{gs}$ gives a very good characterization of the normal and superradiant regions.

## 5 Entanglement

The correlation between radiation and matter also exhibits a distinctive behaviour around the separatrix. To this end we calculate the entropy of entanglement, i.e., the Von Neumann entropy of the reduced density matrix operator; this quantity measures the entanglement between the matter and radiation subsystems.

Figure 10 shows the results for $\chi = 0.01$ and $N_a = 1, 2, 4$ in (a), (b) and (c) respectively. We observe how the blue region, which corresponds to the least significant entanglement, grows in the $\mu_{23}$ direction as the number of particles increases, remaining constant in the $\mu_{12}$ direction. We can also note that as $N_a$ increases, the region of largest entanglement becomes smoother and flatter. The system becomes less entangled as the number of particles increases.

Figure 11 displays the entropy of entanglement for $N_a = 4$ and the intensity of the external field $\chi = 0.3, 0.6, 1$ in columns (a), (b) and (c) respectively. It can be seen that the less entangled region in blue has been reduced significantly and is reduced even further as the value of $\chi$ increases. It is noted a point which has a very high value; this corresponds to the triple-point that has been reported previously for the $\Xi$-configuration [38]. Here we can observe how the blue region, which corresponds to the least significant entanglement, decreases as $\chi$ increases. It is important to note that the maximum values reached for entropy in each case, i.e., the most significant entanglement, generally decrease as the external force increases.

Although both regions (with greater and lesser entanglement) decrease as $\chi$ increases, the ratio between the regions shows an increase in the area of greatest entanglement. It is

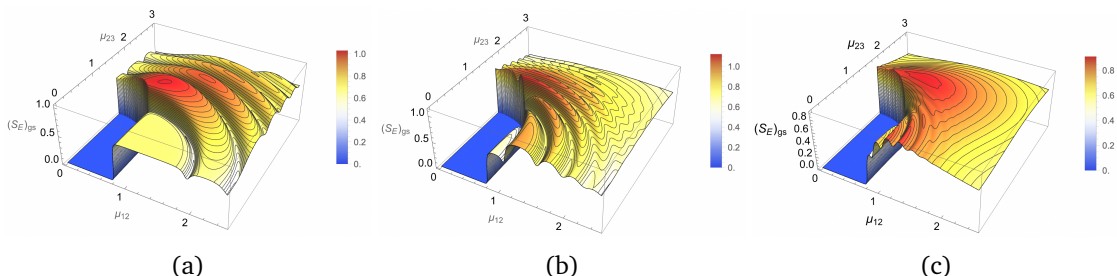

Figure 10: The entropy of entanglement shows the results for $N_a = 1, 2, 4$ with $\chi = 0.01$ in columns (a), (b) and (c) respectively. We observe that the blue region, which corresponds to the less significant entanglement, increases when $N_a$ increases.

concluded that the system becomes increasingly entangled as the external force increases, which coincides with the results obtained with the two-level Dicke model.

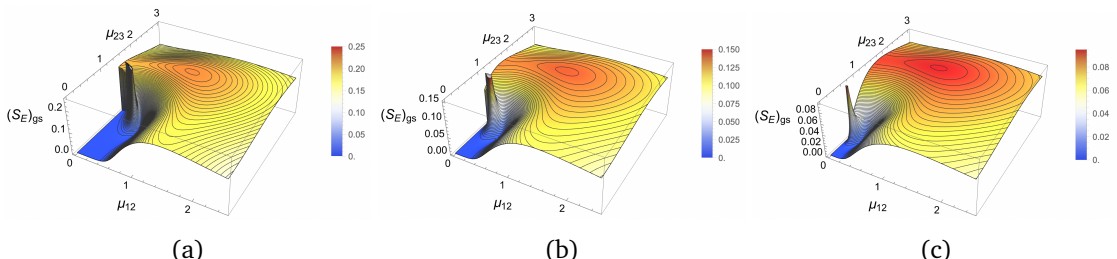

Figure 11: The entropy of entanglement shows the results for $N_a = 4$ and $\chi = 0.3$, 0.6 and 1, in columns (a), (b) and (c), respectively. It is important to take into account the maximum entropy values reached in each case. The ratio between the reduction in the region of least entanglement (blue region) and the reduction in the region of greatest entanglement (red region) generally shows an increase in the area of greatest entanglement. It is concluded that the system becomes increasingly entangled as the external force increases.

# 6 Conclusions

We considered a generalized three-level Tavis-Cummings model in the $\Xi$-configuration and a one-mode electromagnetic field driven by an electromagnetic field. We demonstrated that this model exhibits quantum phase transitions by calculating the expectation value of the number of photons and the population in the lowest and highest energy levels for $N_a = 1, 2, 4$ particles. We observed that the location of the phase transitions are challenging to determine in this form when the number of particles increases. We showed that the Casimir operator of the SU(2) algebras associated with subsystem $1 - 2$ levels can remedy this. This results is shown in agreement with the fidelity between neighbouring states. We observe that the entropy of entanglement, which determines the regions in the parameter space where the system presents entanglement, can be used to determine the normal region; it also allows to distinguish a point in the plots where the behaviour of the observables exhibites an anomaly which is associated to the triple point present in the $\Xi$-configuration [38]. All the plots show that when the intensity of the driven force increases, the normal region decreases, which can be used to observe the quantum phase transition in a system described by the model.

## Acknowledgments

The authors thank to O. Castaños, E. Nahmad-Achar, S. Cordero and V. Romero-Rochín for their valuable comments.

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
