# Peer review of "Quantum phase transition in $\Xi$-configuration Tavis-Cummings model driven by an electromagnetic field for a finite number of atoms"

_SciPost Physics Core_

## Round 1 · Referee Report · Anonymous (Referee 1) · 2025-10-21

Strengths

1- the work contains all mathematical details apt to reproduce the presented derivation. 2- the authors spend some time discussing the computational techniques, allowing to understand better how to reproduce their numerical results. 3- they characterize the system under study with different metrics (the expectation values of operators, entanglement, casimir invariants) providing a complete picture of the model in the parameter space.

Weaknesses

1- In some passages, the abundance of mathematical details is not balanced by as many physical considerations. This might confuse the reader. 2- There are too many plots and in most of them the information on the colorbar (as I infer by the graphs since it is not specified elsewhere) is the same of the one on the z-axis. This is a redundancy. 3- In general, most of Sec. 4 is spent describing the shape in plots without providing any deep physical consideration. 4- Eqs. (8-9) are not used in this work. 5- I think the paper fails to highlight the differences with respect to the case with no drive that was discussed in Ref. [34].

Report

In this work, the authors provide a complete study of the quantum phase transition in the Tavis-Cummings model, in which three-level atom in \Csi configuration are coupled to a common driven EM mode. By computing expectation values of operators in the ground state, the authors find that they are not enough to characterize the full phase transition, inducing the need to consider Casimir invariants. The phase transition is further characterized using the fidelity between neighboring states and the entropy of entanglement between the matter and the radiation degrees of freedom.

While potentially interesting, I would ask the author to made the following changes in order for it to be considered for publication.

Requested changes

1- I would simplify Section 2: I would either say at the beginning that \gamma is arbitrary, or just set it to zero already in the beginning to simplify; I would explain better why the choice of that specific value of \alpha; I would omit Eqs. (8-9). 2- Maybe I would explain in a simpler way the procedure to get the ground state. In the end, since there is a drive the number of excitations is not a good quantum number and so one has to enlarge the Hilbert space to find the GS up to a certain precision. On other hand, in its current form this part seems unnecessarily complicated. 3- In commenting plots, I would comment a bit better the analogy with the two-level TC model. In particular, it seems to me that what is discussed on lines 150-154 is due to the resemblace of the system to a two-level system. 4- In general, I would omit most of the plot and just keep the relevant ones i.e., the ones in which a clear difference is noticed. For instance, I would keep Fig. 5 as it shows that this expectation value fails to capture the transition. 5-I would like to have a clarification on line 260. What this ratio is and how it is computed is not clear at all from the current text. 6-I would highlight better the differences and analogies with the no-drive case, already explored in Ref. [38].

Recommendation

Ask for major revision

---

## Editorial Decision

awaiting_resubmission